Journal of
open psychology data

# Data from the Program for International Student Assessment Young Adult Follow-up Study (PISA YAFS): 2012–2016

DATA PAPER

**DAVID KASTBERG**

**SAIDA MAMEDOVA** 

**SAMANTHA BURG**

*Author affiliations can be found in the back matter of this article

]u[ ubiquity press

## ABSTRACT

The Program for International Student Assessment Young Adult Follow-up Study (PISA YAFS) was conducted in the United States in 2016 with young adults (age 19) who participated in PISA 2012 when they were in high school (age 15). PISA YAFS was designed to measure the relationship between performance on PISA 2012 and subsequent outcomes (education, employment, etc.) as well as skills. These skills were assessed in an online assessment of literacy, numeracy, and problem-solving skills called Education and Skills Online (ESO). ESO was developed to provide individual-level results linked to the Program for International Assessment of Adult Competencies (PIAAC).

**CORRESPONDING AUTHOR:**

**David Kastberg**

Westat, US

DavidKastberg@westat.com

**KEYWORDS:**
Achievement; assessment; data use; databases; employment; large-scale assessment; literacy; performance research design; response rates; secondary education; student assessment; student outcomes

**TO CITE THIS ARTICLE:**
Kastberg, D., Mamedova, S., & Burg, S. (2023). Data from the Program for International Student Assessment Young Adult Follow-up Study (PISA YAFS): 2012–2016. *Journal of Open Psychology Data,* 11: 8, pp. 1–7. DOI: https://doi.org/10.5334/jopd.82

# (1) BACKGROUND

The Program for International Student Assessment Young Adult Follow-up Study (PISA YAFS) was conducted in the United States in 2016 with young adults, approximately 19 years old, who had participated in PISA 2012 when they were in high school, at approximately age 15. The study was designed to measure the relationship between performance on PISA 2012 and subsequent outcomes, including education and employment, as well as skills. These skills were measured in an online assessment of reading literacy, numeracy, and problem-solving skills called Education and Skills Online (ESO), which was originally developed to provide individual-level results linked to the Program for International Assessment of Adult Competencies (PIAAC).

PISA YAFS contributes to research on youth transitions conducted in the United States and internationally. Below, we explore the importance of youth transitions research, discuss the factors that shape this research, examine the international research that has used the PISA and PIAAC datasets, and discuss the findings that have emerged from this research.

Youth transitions are key stages of development that occur when youth move between school levels or leave school. Some of the most commonly studied youth transitions are those that occur at the beginning of adolescence and the beginning of young adulthood. The transition into young adulthood has received particular attention in recent years because of several well-documented cultural and economic shifts. For instance, over the past two decades, there has been an upward trend in postsecondary enrollment and attainment, which has delayed workforce entry for some young adults (Chen et al., 2017; Furstenberg, 2010). At the same time, the costs of higher education have risen dramatically (Horn and Paslov, 2014), leaving increasing numbers of young adults with substantial college debt. Together with an uncertain economy, this debt could dampen young adults' pursuit of future educational, employment, or family goals (College Board, 2017; Settersen and Ray, 2010; Woo and Shaw, 2015). Moreover, these traditional aspects of the transition into young adulthood, such as independent living, marriage, and building a family, have been increasingly delayed by the upward trend in postsecondary enrollment and attainment (Furstenberg, 2010; Settersen and Ray, 2010). Thus, it is important to understand the competencies that young adults possess and the choices they make at this stage, including their relationship to educational, employment, or family goals (College Board, 2017; Settersen and Ray, 2010; Woo and Shaw, 2015).

Recently, at least five other countries have conducted studies following the PISA participants to examine the association between their PISA performance as 15-year-olds and their outcomes as young adults (Borgonovi et al., 2017). In Australia, Switzerland, and Uruguay, these studies were conducted longitudinally, either linking PISA students to ongoing administrative records or using periodic follow-up surveys without skills assessment components. In Denmark, the Ministry of Education administered the PIAAC assessment to a sample of students 12 years after they had initially taken PISA 2000. In Canada, the PISA reading assessment was readministered to a sample of PISA 2000 students 9 years after it was initially administered to them as 15-year-olds.

These studies consistently found strong associations between PISA performance and participants' outcomes as young adults, validating PISA as a potential predictor of students' future success. For example, in Switzerland, PISA 2000 reading performance was positively associated with high school completion and college entry, and in Uruguay, PISA 2003 and 2006 mathematics performance was positively associated with completing high school and negatively associated with dropping out of school (Borgonovi et al., 2017). In Canada, poor PISA 2000 performance was associated with a higher risk of poor labor market outcomes and lower uptake of postsecondary education (OECD, 2010).

In Denmark, the nature of the study allowed it to examine the link between competencies across time (Rosdahl, 2014). This study showed that the higher the PISA 2000 reading score, the higher the probability that the participant was in the top third of performers on the PIAAC literacy scale. However, it also noted certain mitigating factors that were associated with either an increase or decrease in competencies. For example, those participants whose parents were better educated, who themselves had received further education, or who had not had disruptions (e.g., illness or unemployment) that necessitated social welfare payments increased their relative competency level in PIAAC compared to those at similar initial competency levels in PISA 2000.

The study in Australia also found that increased learning opportunities mitigated the relationship between higher socioeconomic status and stronger performance (Borgonovi et al., 2017). Similarly, a cross-sectional study that compared PISA 2000 students with their comparable age cohort in PIAAC 2012 suggested that the negative effects of socioeconomic status on the decline of competencies were more pronounced among the lowest performers than the highest performers (OECD, 2017).[1]

These studies indicate that the PISA results can be used as a lens through which to view youth transitions and show how international data can be used at the national level, without a comparative aspect, to answer research questions of interest. The PISA YAFS study extends this body of international and domestic research that has followed students over time to better understand what predicts and supports their success as young adults.

# (2) METHODS

## 2.1 STUDY DESIGN, MATERIALS, AND SURVEY INSTRUMENTS

PISA YAFS measures both cognitive and noncognitive constructs related to adult-life preparedness, skill use, and achievement in young adulthood. The PISA YAFS data come from two online instruments: the ESO assessment and background questionnaire and an additional module, called the Learning and Educational Career Questionnaire (LEQ), added to supplement the ESO's background questionnaire (Exhibit 1).

The ESO background questionnaire includes a core set of items and multiple noncognitive modules administered to all respondents in addition to the assessment. The noncognitive modules measure self-efficacy related to job seeking, life satisfaction and general affect, and vocational interests. The background questionnaire includes questions about respondents' sex, race/ethnicity, nativity, and language at home, which are used to analyze the PISA YAFS results for different subgroups. However, the background questionnaire did not collect information on current educational status and activities and could not be edited to add content, so the LEQ module was developed. The LEQ module, using questions from the PIAAC background questionnaire, gathered information on current education study status (participation, level of degree, and area of study), formal education activities, and informal learning activities in the 12 months preceding the study.

The ESO assessment was originally developed to provide individual-level results that were linked to the Program for International Assessment of Adult Competencies (PIAAC) and could be obtained independently of PIAAC's decennial administration. The PIAAC (and ESO) framework specifies three content domains: literacy, numeracy, and problem solving in technology-rich environments. Literacy and numeracy were selected as the focus in PISA YAFS because these are the two subjects in the ESO assessment that overlap with the content assessed in PISA 2012.

## 2.2 TIME AND LOCATION OF DATA COLLECTION

The PISA YAFS data were collected in March–July 2016 in the United States. At the start of the PISA YAFS data collection, some 3.5 years after the PISA 2012 assessment, the PISA 2012 students were between the ages of 18 years and 8 months and 19 years and 7 months. The PISA YAFS data were collected online via a secure web survey that was linked to the ESO. Data were extracted from the website and imported into the data management software, where validation checks were performed and data files were prepared.

## 2.3 SAMPLING, SAMPLE, AND DATA COLLECTION

When the PISA 2012 data were collected in October–November 2012 in the United States, the participating students were between the ages of 15 years and 3

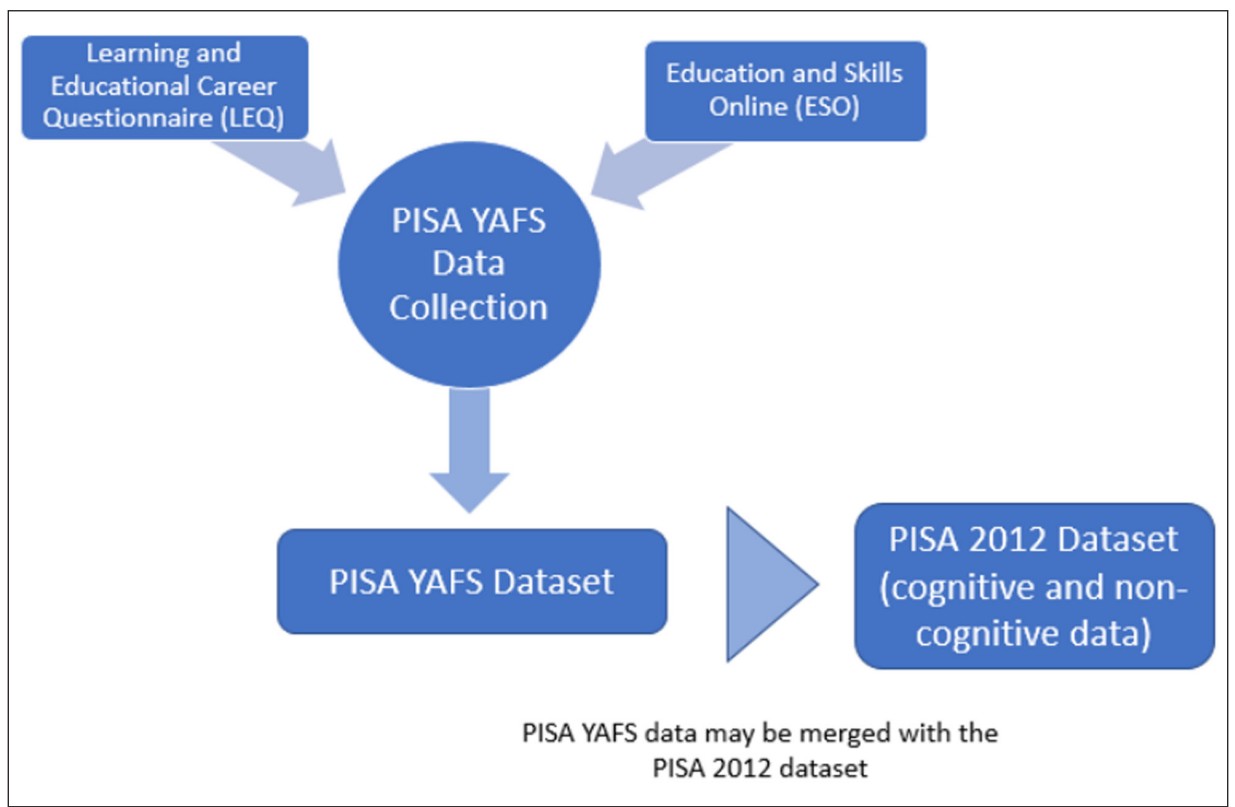

**Exhibit 1** Components of PISA YAFS.

months and 16 years and 2 months. Thus, the PISA YAFS starting sample was the pool of 15-year-old students who took the PISA 2012 mathematics, reading, and science assessments and completed a Student Information Form (SIF) that include their contact information. These students were contacted periodically over the next 4 years through an extensive tracing and recruitment effort to maximize their participation in PISA YAFS, which was voluntary. Nearly 5,000 students took the PISA 2012 mathematics, reading, and science assessments, and 93 percent completed the SIF, resulting in a PISA YAFS starting sample of 4,612 students.

The PISA YAFS data collection involved three main phases: (1) tracing, or the process associated with validating participation-provided contact information and locating the participants; (2) recruitment for participation in PISA YAFS, which involved inviting participants to register for the study website, then establishing and maintaining lines of communication; and (3) data collection, when participants were asked to complete the survey.

The PISA YAFS final sample included 2,318 students, which represented 50 percent of the starting sample. The primary reason for nonparticipation was a lack or loss of contact with the students after they completed the SIF, sometime during the tracing and recruitment effort.

Because of the level of nonparticipation in the final sample, nonresponse bias analyses were undertaken to identify any systematic differences between participants and nonparticipants. The study then made weighting adjustments to reduce any bias detected. As a result, the participants and nonparticipants (or the starting and final samples) are considered comparable, and the results reported are weighted estimates for the PISA YAFS population.

Because the original PISA 2012 sample was nationally representative of the 15-year-old population, and because of the nonresponse bias adjustments in the current study, the PISA YAFS population is nationally representative of individuals who were approximately 19 years old in 2016 and who were 15-year-old students in 2012.

## 2.4 EXISTING USE OF DATA

As of August 2022, only one publication had been released that used the PISA YAFS dataset:

Mamedova, S., Stephens, M., Liao, Y., Sennett, J., & Sirma, P. (2021). *2012–2016 Program for International Student Assessment Young Adult Follow-up Study (PISA YAFS): How Reading and Mathematics Performance at Age 15 Relate to Literacy and Numeracy Skills and Education, Workforce, and Life Outcomes at Age 19* (NCES 2021–029). U.S. Department of Education, National Center for Education Statistics. Washington, DC. Retrieved September 2, 2022, from https://nces.ed.gov/pubsearch/pubsinfo.asp?pubid=2021029.

# (3) DATASET DESCRIPTION AND ACCESS

## 3.1 REPOSITORY LOCATION, LANGUAGE, LICENSE, PUBLICATION DATE

The PISA YAFS data are available as a public-use file and can be downloaded from the NCES website at https://nces.ed.gov/pubsearch/pubsinfo.asp?pubid=2021022. If a data user also wants to use the PISA 2012 data for analysis, they can be downloaded as a separate file from the NCES website at https://nces.ed.gov/pubsearch/pubsinfo.asp?pubid=2014028 and then merged with the PISA YAFS data. The PISA YAFS dataset, published in December 2021, is available in American English with no license requirement.

## 3.2 DESCRIPTION AND CONTENTS OF DATA FILE

The PISA YAFS data are located in an ASCII file named PISA_YAFS2016_Data.dat. This file contains the following:

- questionnaire items, derived variables, and index scores based on
  - the Learning and Educational Career Questionnaire (LEQ),
  - the Education and Skills Online (ESO) core questionnaire, and
  - ESO cognitive and noncognitive items;
- plausible values for the literacy and numeracy scales from the ESO assessment; and
- student nonresponse-adjusted PISA sampling weights and replicate weights.

There are 2,318 cases in the PISA YAFS data file. Since the data are for respondents who took PISA in 2012, each record contains identification variables that enable the user to merge the data with the PISA 2012 student data (using STIDSTD) and with the PISA 2012 school data (using SCHOOLID).

## 3.3 FORMAT NAMES, CODEBOOKS

See Table 1 below for an overview of the supporting documentation available in the PISA YAFS database.

## 3.4 NOTES FOR ANALYSIS

Three aspects of the PISA YAFS design need careful attention in any analysis. The first stems from the sample design. The use of sampling weights is necessary for the computation of statistically sound, nationally representative estimates because the respondents had known, but unequal, probabilities of selection. Sampling weights also adjust for the probabilities of selection to account for nonresponse. Thus, to generalize to the population, sampled analyses will need to apply the sampling weights provided in the file.

The second aspect to be considered also stems from the sampling design and involves the calculation

 

| DOCUMENTATION TYPE | FILE NAME | FUNCTION |
|---|---|---|
| Syntax files | | |
| SPSS | PISA_YAFS2016_SPSS.SPS | Reads the ASCII file into SPSS |
| SAS | PISA_YAFS2016_SAS.SAS | Reads the ASCII file into SAS |
| Codebook | PISA_YAFS2016_Codebook.HTML | Provides information on variable names, variable locations, format information, variable labels, questionnaire text, values, and frequencies |
| "Read me" document | PISAYAFS2016 README_PUBLIC_USE.doc | Lists the file names associated with the public-use dataset |
| Quick-analysis guide | PISA_YAFS2016 QuickGuide.doc | Lists the public-use data file contents, how to create working files, and a data-use agreement |

**Table 1** Overview of supporting documentation in the PISA YAFS database.

of standard errors. Since the sample design is complex (a two-stage, stratified cluster design), most software packages, operating on the assumption of a simple random sample, will produce biased estimates of standard errors. To produce unbiased estimates of the standard error, the analysis undertaken needs to use the replicate weights contained in the file.

The third aspect arises from the design of the PISA YAFS assessment, which requires the use of plausible values in an analysis that includes estimates of proficiency. While the data include individual scores for respondents on ESO cognitive items, in PISA YAFS, as in many national and international assessments, respondents are not administered every assessment item. Instead, respondents answer a subset of the assessment items while their performance on other items is imputed. The results of individual respondents are aggregated to produce a distribution of scores, operationalized through 10 plausible values. This distribution represents a range of abilities for a group of students (e.g., U.S. females), rather than an individual test score result. For analysis purposes, the PISA YAFS dataset includes a set of 10 plausible values for each of the PISA YAFS scales (literacy, numeracy, and problem solving in technology-rich environments). For an analysis undertaken within any of the PISA YAFS assessment domains, all 10 plausible values must be used to calculate the estimate and the standard error.

### 3.5 TOOLS FOR ANALYSIS

Given the complexity of the sampling and assessment design of PISA YAFS, several software packages were adapted to incorporate sampling weight, replicate weights, and plausible values to produce valid estimates. These include EdSurvey (https://www.air.org/project/nces-data-r-project-edsurvey), WesVar (https://www.westat.com/wesvar/), and the International Association for the Evaluation of Educational Achievement (IEA) International Database Analyzer (IDB Analyzer) (https://www.iea.nl/data-tools/tools#section-308),

which produces code for SAS, SPSS, and, in the latest version (5.0), R software packages. For more detail, see the PISA YAFS technical report (https://nces.ed.gov/pubs2021/2021020.pdf).

## (4) REUSE POTENTIAL

PISA YAFS allows research into the characteristics, cognitive skills, and other life outcomes of young adults as they transition from high school to postsecondary life. The initial analysis showed generally strong, positive relationships between individuals' reading and mathematics performance at age 15 and their literacy skills, numeracy skills, and educational trajectories 4 years later. However, it also showed that these relationships are not universal for all students, highlighting students who were in the poorest schools at age 15 as an at-risk group for skill loss (Mamedova et al., 2021). Building upon these findings, potential further explorations include analyzing the data for different population groups, such as racial/ethnic groups that exhibited different skills trajectories from those of the overall 19-year-old population (Mamedova et al., 2021). Researchers could also consider modeling other contributing factors to these trajectories while controlling for relevant background variables. Furthermore, studying the population of young adults who moved from the low-performance category at age 15 to the high-performance category at age 19 might yield some valuable lessons.

Moving outside of the framework set in the previously published report, an expanded analysis could incorporate more variables from the PISA 2012 dataset to study the employment and education paths of students, shedding light on societal and family structures. By including more of the PISA 2012 variables, researchers could examine the potential reciprocal relationships between measures of student affect and self-concept, on the one hand, and literacy and numeracy achievement, on the other. This analysis would replicate previous research

Kastberg et al. *Journal of Open Psychology Data* DOI: 10.5334/jopd.82

using the PISA 2000 and 2003 datasets (Williams et al., 2005; Williams & Williams, 2010), which showed close relationships between these factors in both reading and mathematics.

## SPECIAL COLLECTION

Data for Psychological Research in the Educational Field Editors of the Special Collection: Sonja Bayer, Katarina Blask, Timo Gnambs, Malte Jansen, Débora Maehler, Alexia Meyermann, Claudia Neuendorf (alphabetic order).

## NOTE

[1] In the referenced OECD study, socioeconomic status was measured by parents' educational attainment and the number of books in the home when students were 15 to 16 years old.

## ACKNOWLEDGEMENTS

The authors wish to thank the young adults who participated in PISA YAFS. Without their cooperation and willingness to share information about themselves, this study would not be possible. The authors also wish to thank Daniel McGrath and Dana Kelly at the National Center for Education Statistics (NCES) for their vision and work to make PISA YAFS possible. The authors would also like to thank Patrick Gonzales, the former project officer for PISA at NCES, who oversaw the PISA YAFS study from its inception to May 2020.

## FUNDING INFORMATION

This project has been funded through a contract with the National Center for Education Statistics (NCES) of the U.S. Department of Education [2013 ED-IES-13-C-0006].

## COMPETING INTERESTS

The authors have no competing interests to declare.

## AUTHOR CONTRIBUTIONS

The authors wish to recognize the contributions of several people who were integral in making the PISA YAFS study successful: Gordon Murray, David Ferraro, Shep Roey, Carlos Arieira, Robert Perkins, and Loydis Cummings at Westat; Irwin Kirsch, Ann Kennedy, and Kentaro Yamamoto at Educational Testing Service (ETS); and Maria Stephens, Yuqi Liao, Josh Sennett, and Paul Sirma at the American Institutes for Research (AIR).

## AUTHOR AFFILIATIONS

**David Kastberg**
Westat, US

**Saida Mamedova** (ID) orcid.org/0000-0002-4007-0340
American Institutes for Research, US

**Samantha Burg**
National Center for Education Statistics, US

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

## PEER REVIEW COMMENTS

*Journal of Open Psychology Data* has blind peer review, which is unblinded upon article acceptance. The editorial history of this article can be downloaded here:

- **PR File 1.** Peer Review History. DOI: https://doi.org/10.5334/jopd.82.pr1

**TO CITE THIS ARTICLE:**
Kastberg, D., Mamedova, S., & Burg, S. (2023). Data from the Program for International Student Assessment Young Adult Follow-up Study (PISA YAFS): 2012–2016. *Journal of Open Psychology Data,* 11: 8, pp. 1–7. DOI: https://doi.org/10.5334/jopd.82

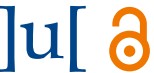