## [Peer Review History. · Journal of Open Psychology Data]

Reviewer 1:

Thank you for the possibility to review the article "Data from the Program for International Student Assessment Young Adult Follow-up Study (PISA YAFS): 2012-2016". The presented data stem from a follow-up survey of PISA 2012 administered in 2016 in the United States. It is called PISA YAFS and was designed to study the development of students as they transition into young adulthood. In my opinion, the data is of high quality and very useful for researchers to conduct further investigations. The data is freely available for download and well documented. In addition, several control files to import the data are provided on the website. Furthermore, the authors mention helpful tools and instructions for data analysis considering the sampling structure and weights in the article. In general, the data and data documentation described in the article meet the requirements of the journal. However, I have some suggestions that might help improve the paper's quality before publication.

- Why do not you use the subheadings provided in the JOPD paper template? I think the article would benefit from implementing this suggested structure, as it would make it easier for readers to find the information they are looking for.

Authors: Added JOPD subheadings.

- Please check for the correct use of APA citation style in the text and the reference list.

Authors: Checked, edited.

- The link to the data files leads to the description of the files, not the "publications/data products" page where the files can be downloaded. Perhaps you could also provide the link to the publications/data products page in the text.

Authors: Changed the link to go directly to the publications/data products page to be more accurate.

Minor comments:

- Pg. 1: Missing dot at the end of the sentence: "One of the most commonly studied youth transitions includes those that occur at the beginning of adolescence and at the beginning of young adulthood "

Authors: Revised

- Pg. 1: Missing comma: "At the same time, ..."

Authors: Revised

- Pg. 8: Missing information in the brackets: "...and in the latest version () R software packages."

Authors: R version added.

Reviewer 2:

Dear Authors,

Thank you for the opportunity to get an insight into the interesting PISA Follow-up Study.

An exciting data set with analysis potential for looking at the further development of PISA students in the United States. As already implemented in other countries, the study builds on the PISA data and surveys the respondents at a later stage (when they are about 19 years old).

For me, the presentation of the study was comprehensible. Nevertheless, the text could benefit from being more structured in various places (marked in the document attached).

In the description of the background, for example, it would be helpful to differentiate more clearly between what the given social conditions are and what the specific potential of the data presented is.

Authors: Added a sentence in the beginning of the background (second paragraph) to clarify what information will be provided in the paragraphs following.

It would also be interesting to know whether there are other studies with a longitudinal design in the United States and what distinguishes these data in comparison.

Authors: Since background had word limits and since the journal seems to be targeted to the international audience, we have omitted the discussion on longitudinal studies in the United States. However, that is available in the report (pp.2-3) : <https://nces.ed.gov/pubs2021/2021029.pdf>.

In addition, it would be helpful, especially when presenting the data set, if the contents were presented graphically and not just described in continuous text. Here, for example, tables or illustrations could enhance the text.

Authors: Added bullet points and transferred one section into a tabular format.

Overall, I would have liked a little more information on sampling and study design, but this can certainly be found in the general PISA documentation (maybe these could be linked?). The potential for further use is visible, but could be structured and elaborated more clearly.

Authors: Added more information on the sampling and data collection.

I am in favour of a revision and think that the paper could provides useful insights into a freely available dataset.